# Deficits in Key Brain Network for Social Interaction in Individuals with Schizophrenia

**DOI:** 10.3390/brainsci13101403

**Published:** 2023-09-30

**Authors:** Yiwen Wu, Hongyan Wang, Chuoran Li, Chen Zhang, Qingfeng Li, Yang Shao, Zhi Yang, Chunbo Li, Qing Fan

**Affiliations:** 1Shanghai Mental Health Center, Shanghai Jiao Tong University School of Medicine, Shanghai 200030, China; wuyiwenhsd@163.com (Y.W.); wanghongyanha@163.com (H.W.);; 2Beijing Key Laboratory of Mental Disorders, National Clinical Research Center for Mental Disorders and National Center for Mental Disorders, Beijing Anding Hospital, Capital Medical University, Beijing 100088, China; 3Advanced Innovation Center for Human Brain Protection, Capital Medical University, Beijing 100069, China; 4Shanghai Key Laboratory of Psychotic Disorders, Shanghai 200030, China; 5Institute of Psychology and Behavioral Science, Shanghai Jiao Tong University, Shanghai 200030, China; 6Mental Health Branch, China Hospital Development Institute, Shanghai Jiao Tong University, Shanghai 200030, China

**Keywords:** schizophrenia, resting-state fMRI, functional connectivity, reward network, emotional salience network

## Abstract

Individuals with schizophrenia (SZ) show impairment in social functioning. The reward network and the emotional salience network are considered to play important roles in social interaction. The current study investigated alterations in the resting-state (rs-) amplitude of low-frequency fluctuation (ALFF), fractional ALFF (fALFF), regional homogeneity (ReHo) and functional connectivity (fc) in the reward network and the emotional salience network in SZ patients. MRI scans were collected from 60 subjects, including 30 SZ patients and 30 matched healthy controls. SZ symptoms were measured using the Positive and Negative Syndrome Scale (PANSS). We analyzed the ALFF, fALFF and ReHo in key brain regions in the reward network and emotional salience network as well as rs-fc among the bilateral amygdala, lateral orbitofrontal cortex (OFC), medial OFC and insula between groups. The SZ patients demonstrated increased ALFF in the right caudate and right putamen, increased fALFF and ReHo in the bilateral caudate, putamen and pallidum, along with decreased fALFF in the bilateral insula. Additionally, reduced rs-fc was found between the right lateral OFC and the left amygdala, which simultaneously belong to the reward network and the emotional salience network. These findings highlight the association between impaired social functioning in SZ patients and aberrant resting-state ALFF, fALFF, ReHo and fc. Future studies are needed to conduct network-based statistical analysis and task-state fMRI, reflecting live social interaction to advance our understanding of the mechanism of social interaction deficits in SZ.

## 1. Introduction

Schizophrenia (SZ) is a chronic psychiatric disorder with a substantial disease burden [1]. Positive symptoms and negative symptoms are two primary dimensions of SZ symptoms [2], and both sets of symptoms contribute to impaired social functioning in SZ patients [3]. Positive symptoms, including hallucinations, delusions and disorganization, would erode patients’ trust in others [4] and affect their response to social rewards [5]. While negative symptoms, characterized by the loss of motivation, goal-directed behaviors and affects, would exacerbate defeatist performance beliefs [6], impede daily functioning and hinder recovery [7]. A longitudinal study spanning eight years demonstrated a significant association between negative symptoms and diminished social functioning in SZ patients [8]. Improving the social functioning of SZ patients is considered a priority target for SZ rehabilitation [9].

The decline in social functioning among individuals with SZ is closely related to deficits in social cognition [10,11]. Representative domains of social cognition encompass the theory of mind, social perception, emotion processing and attributional style/bias [12]. In SZ patients, the deficits in social cognition have been widely reported [13], for example, the hostile attributions in SZ contributed to poor social functioning [14], and an impaired theory of mind was associated with negative symptoms [15].

Neuroimaging studies on healthy adults have found that the reward network and the emotional salience network play critical roles in social cognition and were identified as key brain networks in the theory of “Social Brain” [16]. The reward network includes the dorsal and ventral striatum, ventral tegmental area, substantia nigra, amygdala, anterior cingulate cortex, insula, orbitofrontal cortex (OFC) and medial prefrontal cortex(MPFC) [17]. Notably, the striatum and OFC are widely recognized in neuroimaging research as central to reward processing [18], because both of them are involved in reward-based decision-making, reward learning and approach motivation in social interactions, particularly in response to positive social feedback [19]. Neurons in the amygdala were also shown to discriminate social cues and take part in social decision-making [20]. The emotional salience network comprises brain regions associated with emotional salience, such as the dorsal anterior cingulate cortex (dACC) and insular cortices [21] as well as brain regions involved in emotional regulation like the temporal pole, middle temporal gyrus [22], amygdala and parietal cortex [23]. Specifically, the dACC is sensitive to events that deviate from expectations and will be activated when mismatch happens during social interaction [24], while the insula, integral in interoceptive and emotional processing, exhibits sensitivity to emotional pain like social rejection [25]. In addition to memory retrieval and language processing, the temporal pole and the middle temporal gyrus also contribute to social and emotional processes, including face recognition and theory of mind [26].

Previous neuroimaging studies have shown varying degrees of deficits in the reward network and the emotional salience network among SZ patients [27,28]. Within the reward network, SZ patients showed aberrant reward-related striatal signaling [19], and an impaired representation of value-relevant OFC function in response to both social [29,30] and nonsocial feedback [31]. An increased resting-state functional connectivity between the amygdala, hippocampus and OFC was also found in SZ patients with paranoia [32]. In the emotional salience network, recent studies revealed that individuals with SZ manifest aberrantly increased functional connectivity between the salience network and the right inferior and middle temporal gyrus in the resting-state [33]. Compared to healthy controls, SZ patients demonstrated an overall reduction in insula-to-whole-brain functional connectivity [32], coupled with diminished activity in the insula and anterior cingulate cortex during the ambivalence task [28].

In resting-state functional magnetic resonance imaging (rs-fMRI) studies, two types of neural imaging markers are commonly used to reflect the local brain functional changes: (1) the amplitude of low-frequency (0.01–0.08 Hz) fluctuation (ALFF) [34] or fractional amplitude of low-frequency fluctuation (fALFF, i.e., the ratio of power spectrum of low-frequency to that of the entire frequency range [35]) and (2) regional homogeneity (ReHo, which calculates the consistency of time series between each voxel and adjacent voxels [36]). ReHo and functional connectivity (fc) analyses focus on the similarities of intra- and inter-regional time series, respectively, while ALFF and fALFF measures the amplitude of regional activity [37]. These methods demonstrate satisfactory stability and reliability in assessing spontaneous neural activity in both SZ patients and healthy controls (HC) [37].

In schizophrenia, robust abnormalities in resting-state brain activity have been observed. Shao et al. reported increased ALFF in the left caudate and decreased ALFF in the bilateral posterior cingulate cortex/precuneus (PCC/PCu) in early-course SZ patients compared to healthy controls (HC) [38]. A meta-analysis by Qiu et al. reviewed spontaneous brain activity alterations in SZ patients, including abnormalities in the putamen, lateral globus pallidus, insula, cerebellum and PCC [37]. Another meta-analysis found that chronic SZ group showed increased ReHo in bilateral superior frontal gyrus (SFG) and right insula as well as decreased ReHo in left medial frontal gyrus and left ACC [39].

However, previous studies mostly conducted a general observation of abnormalities from a whole-brain perspective, without delving into the specific brain regions associated with social interaction networks. Therefore, the current study aims to investigate the alterations in ALFF, fALFF, ReHo and fc at rest within and between two socially relevant networks—the reward network and the emotional salience network. Additionally, this study seeks to explore the correlation between spontaneous neural activity and current symptom severity as well as social functioning in SZ patients. This study hypothesized that individuals with SZ would demonstrate abnormal ALFF, fALFF and ReHo in key components of the reward network and emotional salience network, and show deficits in functional connections within and between them. Furthermore, the study expected that altered resting-state ALFF, fALFF, ReHo and fc would correlate with symptom severity and social functioning in SZ patients.

## 2. Materials and Methods

### 2.1. Subjects

Thirty SZ patients were recruited from an interventional study at Shanghai Mental Health Center (clinical trial registration number: ChiCTR-IOR-15006678). The baseline data of the interventional study were incorporated into the current study.

The inclusion criteria for SZ patients were: (a) between 18–55 years old; (b) confirmed diagnosis of schizophrenia according to the Diagnostic and Statistical Manual of Mental Disorders, Fifth Edition (DSM-5); (c) the symptoms were stabilized after treatment with second-generation antipsychotics, and the medication types remained relatively stable for eight weeks; (d) an education level beyond primary school (⩾5 years of education); and (e) having read and signed informed consent explaining the purpose and procedures of the study, indicating that they have understood the purpose of the required procedure and volunteered to participate in the study.

Exclusion criteria for the SZ group were: (a) received modified electroconvulsive therapy or systemic psychotherapy within 6 months before screening; (b) alcohol or drug abuse (defined by DSM-5 [40]) within 6 months before screening; (c) developmental disability, (d) display of excited, impulsive or aggressive behavior; (e) have suicidal behavior or existing suicidal tendencies; (f) a history of cardiovascular, pulmonary, renal, hepatic, gastrointestinal, neurological, hematological, endocrine or metabolic diseases; (g) insufficient visual or auditory function to complete the test; and (h) any other condition preventing participants from completing the test.

Thirty age-, gender- and education-matched healthy controls (HC) were recruited through advertisements. Additional exclusion criteria for the healthy controls included: (a) having a first-degree relative with major psychiatric illnesses; (b) alcohol or drug abuse (defined by DSM-5 [40]); (c) a history of cardiovascular, pulmonary, renal, hepatic, gastrointestinal, neurological, hematological, endocrine or metabolic diseases; (d) insufficient visual or auditory function to complete the test; and (e) any other condition preventing participants from completing the test.

All patients received stable antipsychotic medication treatment (including aripiprazole (n = 4), olanzapine (n = 9), quetiapine (n = 1), risperidone (n = 2), clozapine (n = 12) and amisulpride (n = 2)). Considering that the medication types varied among patients, we calculated the mean olanzapine equivalents (OL eq.) using the equivalence scales [41] (see Table 2).

### 2.2. Assessments

#### 2.2.1. Schizophrenia Symptom Assessment

The schizophrenia symptoms were measured using the psychiatrist–rated Positive and Negative Syndrome Scale (PANSS) [42]. PANSS evaluates the severity of psychosis syndromes, including positive symptoms, negative symptoms and general symptoms, each with 7,7 and 16 sub-items, respectively. Scores are assigned on a 7-point scale of 1–7 points ranging from “none” to “extremely severe”. The total score of PANSS ranges from 30 to 210 points, with higher scores indicating a greater severity of symptoms.

#### 2.2.2. Social Functioning Assessment

The social functioning of SZ patients was indirectly assessed by the Schizophrenia Quality of Life Scale (SQLS) [43]. This self-administered questionnaire captures the perceptions and concerns of individuals with SZ, which contains a set of 30 items comprising 3 dimensions: ‘psychosocial’ (15 items), ‘motivation and energy’ (7 items) and ‘symptoms and side-effects’ (8 items). Higher SQLS scores indicate poorer quality of life for the patient. The SQLS demonstrated satisfactory internal consistency reliability and validity.

### 2.3. MRI Acquisition and Preprocessing

#### 2.3.1. Image Data Acquisition

The T1-weighted images and resting-state fMRI data were acquired using a 3T Siemens Verio MR scanner (Erlangen, Germany) at Shanghai Mental Health Center. T1-weighted images were acquired using multi-band accel factor and magnetization-prepared rapid gradient-echo (MPRAGE) sequence with the following parameters: echo time (TE) = 3.65 ms, repetition time (TR) = 2530 ms, field of view (FOV) = 256 × 256 mm${}^{2}$, flip angle = 7°, voxel size = 1.1 × 1.0 × 7.0 mm${}^{3}$ and thickness = 1 mm, slice number = 192. The resting-state fMRI data were acquired using multi-band accel factor and a gradient-echo Echo Planar Imaging (EPI) sequence with the following parameters: echo time (TE) = 30 ms, repetition time (TR) = 2000 ms, field of view (FOV) = 220 × 220 mm${}^{2}$, flip angle = 77°, voxel size = 3.0 × 3.0 × 3.0 mm${}^{3}$, thickness = 3 mm and slice number = 50.

#### 2.3.2. Image Data Processing

For the T1 MRI images, we first performed quality control using CAT12 to exclude subjects with quality scores of “C+” or lower. Next, we used the recon-all command of FreeSurfer 6.0.0 for brain extraction, tissue segmentation, cortical reconstruction and brain region labeling for each subject’s brain images. The image processing procedures involved skull stripping, normalization, removal of the non-brain structure, brain tissue segmentation and surface reconstruction. Finally, we extracted the thickness, surface area, volume for cortical structures and volume of subcortical structures in different brain regions based on the DK+Aseg parcellation template implemented in FreeSurfer 6.0.0. Finally, the brain images of each subject were nonlinearly aligned to the MNI152 template using the advanced normalization tools (ANTs) alignment toolkit for subsequent analysis of BOLD fMRI images.

For resting-state functional MRI images, we mainly used the AFNI toolkit for processing [44,45]. The first five volumes were removed before motion correction to allow for magnetization equilibrium. After that, head motion was corrected using AFNI’s 3dvolreg. Power’s framewise displacement (FD) was calculated to reflect volume-wise head movement [46], and those whose mean FD > 5 mm were excluded. Assessment metrics that measured the degree of rotation and displacement of the head were generated in all three directions. Then, images of each subject were corrected for temporal layers. The images of the intermediate time points were aligned with the T1-weighted images, and head movement outlier time points in the temporal dimension were detected and interpolated using adjacent time points. Afterward, for each voxel corresponding to the BOLD signal sequence, the mean signal of white matter and the ventricle, as well as the three noise sets of head movement parameters, were regressed off and band-pass filtered in the range of 0.01–0.1 HZ. Next, the mean value of the whole-brain BOLD signal was adjusted to 10,000. Finally, we used the alignment results of the T1-weighted image of each subject to transform the resting-state images into MNI152 space and calculated the functional connectivity between all brain regions, as well as the amplitude of low-frequency fluctuation (ALFF), fractional amplitude of low-frequency fluctuation (fALFF) and regional homogeneity (ReHo) mean values for each brain region according to the DK+Aseg partitioning template [47,48].

#### 2.3.3. Region of Interest

The components of the reward network and emotional salience network have been described in previous studies (see Table 1) [16]. Among them, the bilateral amygdala, lateral OFC, medial OFC and insula belong to both of the two networks, playing roles as vital nodes in brain networks for social interaction. Therefore, we designated these eight brain regions as regions of interest (ROIs).

#### 2.3.4. Statistics

For demographics and clinical data, we conducted independent *t*-tests in SPSS26.0 to compare the age and education years between SZ patients and HC, and performed a chi-squared test to compare gender between groups. For the ALFF, fALFF and ReHo, we used SPSS26.0 to perform an independent *t*-test between the groups. For ROI-to-ROI analyses, we calculated correlation coefficients of the timeseries of the blood-oxygenation-level-dependent (BOLD) signals between ROIs and converted them to z values. Then, we used an independent samples *t*-test to determine altered ROI-based functional connections between SZ patients and HC, and used R for FDR correction of *P* values. We set the significance threshold at $P_{FDR}$ < 0.05. Additionally, we conducted Spearman correlations between the ALFF, fALFF, ReHo, re-fc and the scores of the PANSS and SQLS-P. The medication dosage (i.e., OL eq.) of each patient was used as a covariate of non-interest during correlation analysis.

## 3. Results

### 3.1. Demographics and Clinical Data

Between the SZ group and the HC group, no differences were found in gender, age and education (*P*s > 0.05). The detailed demographic and clinical outcomes are presented in Table 2.

### 3.2. ALFF, fALFF and ReHo Differences of Key Brain Regions between Two Groups

The mean values of altered ALFF, fALFF and ReHo of key brain regions in the reward network and the emotional salience network (see Table 1) were included in the analysis.

#### 3.2.1. ALFF Group Difference

Compared with HC, the SZ group had significantly increased ALFF in the right caudate (*t*(58) = 4.40, $P_{FDR}$ = 0.001, *Cohen’d* = 1.136) and right putamen (*t*(58) = 3.50, $P_{FDR}$ = 0.009, *Cohen’d* = 0.903) (Figure 1, Table 3). No significant correlation was found between ALFF and the PANSS score or the SQLS score after FDR correction.

#### 3.2.2. fALFF Group Difference

The SZ group had increased fALFF in the bilateral caudate (right: (*t*(58) = 2.732, $P_{FDR}$ = 0.038, *Cohen’d* = 0.705; left: (*t*(58) = 2.708, $P_{FDR}$ = 0.038, *Cohen’d* = 0.699), bilateral putamen (right: (*t*(58) = 2.717, $P_{FDR}$ = 0.038, *Cohen’d* = 0.702; left: (*t*(58) = 2.506, $P_{FDR}$ = 0.038, *Cohen’d* = 0.647) and bilateral pallidum (right: (*t*(58) = 2.629, $P_{FDR}$ = 0.038, *Cohen’d* = 0.679, left: (*t*(58) = 2.518, $P_{FDR}$ = 0.038, *Cohen’d* = 0.65) compared with HC. Furthermore, the SZ group showed decreased fALFF in the insula (right: (*t*(58) = −3.082, $P_{FDR}$ = 0.038, *Cohen’d* = 0.796; left: (*t*(58) = −2.608, $P_{FDR}$ = 0.038, *Cohen’d* = 0.673) (Figure 2, Table 4). No significant correlation was found between fALFF and the PANSS score or the SQLS score after FDR correction.

#### 3.2.3. ReHo Group Difference

For ReHo, the SZ group had increased ReHo in the bilateral caudate (right: *t*(58) = 4.048, $P_{FDR}$ = 0.002, *Cohen’d* = 1.045; left: *t*(58) = 3.987, $P_{FDR}$ = 0.002, *Cohen’d* = 1.03), bilateral putamen (right: *t*(58) = 3.601, $P_{FDR}$ = 0.004, *Cohen’d* = 0.93; left: *t*(58) = 3.393, $P_{FDR}$ = 0.006, *Cohen’d* = 0.876) and bilateral pallidum (right: *t*(58) = 2.761, $P_{FDR}$ = 0.030, *Cohen’d* = 0.713; left: *t*(58) = 2.697, $P_{FDR}$ = 0.030, *Cohen’d* = 0.696) compared with HC. (Figure 3, Table 5). No significant correlation was found between ReHo and the PANSS score or the SQLS score after FDR correction.

### 3.3. Altered Resting-State Functional Connections between Reward Network and Emotional Salience Network

In ROI-to-ROI analysis, SZ patients showed significantly decreased rs-fc between the R. lateral orbitofrontal cortex and L. amygdala (*t*(58) = 3.726, $P_{FDR}$ = 0.012, *Cohen’d* = 0.962) (Figure 4). However, rs-fc between the R. lateral orbitofrontal cortex and L. amygdala was not correlated with the PANSS score (Positive: *r* = −0.172, *p* = 0.362; Negative: *r* = 0.115, *p* = 0.544; General: *r* = −0.181, *p* = 0.339; Total: *r* = −0.207, *p* = 0.272) or the SQLS score (*r* = 0.222, *p* = 0.309) after FDR correction. We also compared the rs-fc of other regions of the reward network and the emotional salience network between the SZ group and HC group, but no significant difference was found after FDR correction.

## 4. Discussion

The principal finding revealed the deficits in brain networks associated with social interaction in individuals with schizophrenia. From a global perspective, the current study observed diminished functional connectivity between the reward network and the emotional salience network, reflected in the reduced rs-fc between the right lateral OFC and the left amygdala. From a more concrete view, compared with the HC group, the SZ group showed increased ALFF in the right caudate and right putamen, along with increased fALFF and ReHo in the bilateral caudate, putamen and pallidum, while fALFF in the bilateral insula significantly decreased. In this study, resting-state cerebral perfusion and functional connectivity did not exhibit significant correlations with PANSS scores and SQLS scores.

### 4.1. Altered ALFF, fALFF and ReHo in the SZ Group

In this study, we found increased fALFF and ReHo in the bilateral caudate, putamen and pallidum in the SZ group compared to the HC group, which is consistent with previous studies [37,49,50]. The caudate, putamen and pallidum collectively form the dorsal striatum, a crucial node in the dopamine pathway [51]. In healthy people, studies have found that striatum plays an important role in integrating sensorimotor, cognitive and motivational/emotional information as well as decision-making [52,53]. Functional studies in individuals with schizophrenia have reported increased striatal dopamine neurotransmission and striatal activity [54], while blunted striatal reward signals might contribute to deficits in motivation and hedonism [55]. Recent researchers have also identified structural changes in the dorsal striatum in SZ, including enlarged gray matter volume in the putamen [56,57], as well as genetic variations, like the dopamine D2 receptor short isoform in the caudate, which are associated with schizophrenia risk [58]. Therefore, thw aberrantly increased activations observed in the dorsal striatum of SZ patients may indicate abnormalities in their processing of social rewards.

Compared with the HC group, SZ patients in this study showed decreased fALFF in the insula. Previous studies have reported varying results: one study found lower fALFF in the left insula for deficit schizophrenia, characterized by primary and persistent negative symptoms, compared to non-deficit schizophrenia [59], while another study found increased fALFF in the insula for SZ patients with persistent auditory verbal hallucinations, compared to non-hallucinating SZ patients and HC [60]. These discrepant findings indicate that variations in the primary symptoms of SZ patients may influence their spontaneous neural activity. Neuroimaging studies have highlighted the anterior insula’s pivotal role in emotional experience and social cognition, including empathy [61]. Similar to our findings, previous studies have noted decreased resting-state cerebral blood flow in the insula in SZ patients [62]. Beyond emotional processing, the insula is crucial for discriminating self-generated and external information, suggesting that insula dysfunction may contribute to hallucinations, a cardinal feature of schizophrenia [63,64].

### 4.2. Altered Functional Connectivity in the SZ Group

Decreased orbitofrontal-amygdaloid functional connectivity was frequently found in the SZ group [65,66,67]. In this study, decreased rs-fc between the R. lateral OFC and L. amygdala was also observed. The amygdala is neurochemically and physiologically sensitive to stress [68] and plays an important role in affective salience processing [69]. Amygdala lesions are often considered central to schizophrenia-related psychopathology [67]. The OFC, in turn, exerts inhibitory control over subcortical structures, including the amygdala, and contributes significantly to affective decision-making in humans [70]. Functional impairment in the OFC leads to increased impulsivity [71] and reduced emotional flexibility [72]. Hence, the deficiency in orbitofrontal-amygdaloid connectivity may indicate a diminished capacity in individuals with schizophrenia to make value-based decisions and regulate emotions.

Neuroimaging research has identified the amygdala and OFC as crucial components in the reward network and emotional salience network, which are activated during social interactions [16]. In the reward network, the amygdala not only takes part in nonsocial contexts but is also involved in processing social rewards, such as positive feedback in social interaction [73] and fair cooperation during economic games [74]. The emotional salience network serves as a salience detector when one is included in/excluded from social interaction, directing attention to experiences of emotional engagement (especially the pain of social rejection) (see reference [35] for a review). Engagement in, or the belief of engaging in, live social interaction can increase the activation of the OFC in the emotional salience network [75,76]. Thus, the decreased functional connectivity observed in this study between the emotional salience network (integral to attentional processes for personally relevant or highly salient events) and the reward network (essential for value representation of socially rewarding stimuli) may shed light on why patients with SZ would show suspiciousness and feelings of persecution and apathy during social interactions. Studies using passive observation paradigms and social engagement paradigms have also demonstrated decreased affective processing of social reward in individuals with SZ compared to healthy people (reviewed, for example, in reference [5]). Longitudinal studies are needed to ascertain whether the altered perception of social reward and punishment in SZ patients stems from difficulties in directing attention to themselves and emotional processing, or if the attention deficit arises from aberrant value representation.

### 4.3. The Correlation between ALFF, fALFF, ReHo, rs-fc and Symptom Severity or Social Functioning

No significant correlation between altered ALFF, fALFF, ReHo, rs-fc and the PANSS score or the SQLS score was found in this study, which is contrary to our hypothesis. Similar to our results, a cross-sectional study including different stages of SZ found that, rather than ALFF or ReHo values, only gray matter volume was negatively correlated with total PANSS scores [54]. In Cheon’s study, negative symptom scores showed predominantly stronger positive associations with ALFF in the temporal and frontal brain regions [77]. However, Chen’s research demonstrated that OFC-related re-fc was negatively correlated with excited symptoms in first-episode patients with SZ [78]. Therefore, further evidence on the relationship between spontaneous brain activity and clinical symptoms in SZ patients is required for consensus. Although SQLS is commonly used to measure the life quality of SZ patients and contains the ‘psychosocial’ dimension, it lacks specialization in assessing social functioning [43]. Thus, it may not be sufficiently sensitive to discern varying levels of social functioning, potentially causing the negative results. Future studies should use scales explicitly designed for measuring social functioning, such as the social functioning scale (SFS) [79], to explore the correlation between aberrant social-related brain networks and social functioning deficits in SZ patients.

### 4.4. Limitations and Future Studies

This study has several limitations. Firstly, the sample size was relatively small when considering that various spontaneous brain activity analyses were conducted exploratorily. A larger sample size of the homogenous population is essential for robustly testing neural activities in the social-related brain networks of SZ patients. Secondly, it is worth noting that antipsychotics may potentially influence symptom severity and rs-fc in SZ patients [80]. While efforts were made to control for this by using the olanzapine equivalence scale to assess medication dosage and adjusting for it in data analysis, it is important to acknowledge that the complete medication profile of each patient could not be captured due to the lack of certain medications (e.g., selective serotonin reuptake inhibitors) in the scale. Future studies should endeavor to conduct large randomized controlled trials to establish a more comprehensive equivalence scale. Thirdly, this study did not use directly relevant scales or behavioral tasks to assess individuals’ social functioning. It will be necessary for future studies to use relevant instruments such as the social functioning scale (SFS) [79], behavioral tasks, like the Cyberball task [81], and passive social observation paradigms [82] to explore the direct correlation between aberrant social-related brain networks and the deficits in social functioning in individuals with SZ. Finally, as a resting-state fMRI study, we failed to examine the real-time activation of social-related brain networks during individuals’ social interactions by using tasks or paradigms. Task-state fMRI could be employed in future studies involving patients with SZ to further investigate the changes in society-related brain networks during social interactions. What is more, future studies are also required for further analysis, like network-based statistical analysis (NBS) [83], in order to elucidate the complex characteristics and functional connections of social-related brain networks and to delineate the map of the social brain in individuals with SZ.

## 5. Conclusions

This study identified deficits in key brain regions for social interaction, with increased activations in the bilateral dorsal striatum (including the caudate, putamen and pallidum) and decreased activations in the bilateral insula among individuals with SZ. No significant correlation was observed between this abnormal neural pattern and symptom severity or social functioning in SZ. The decreased orbitofrontal-amygdaloid functional connectivity underscores the abnormal connection within key brain networks (i.e., the reward network and the emotional salience network) for social interaction in SZ. To facilitate our understanding of the interplay within social-interactive brain networks, network-based statistical analysis and task-state fMRI, reflecting real-time social interactions, will be required to elucidate the mechanism behind social skill deficits in SZ. 

## Figures and Tables

**Figure 1 brainsci-13-01403-f001:**
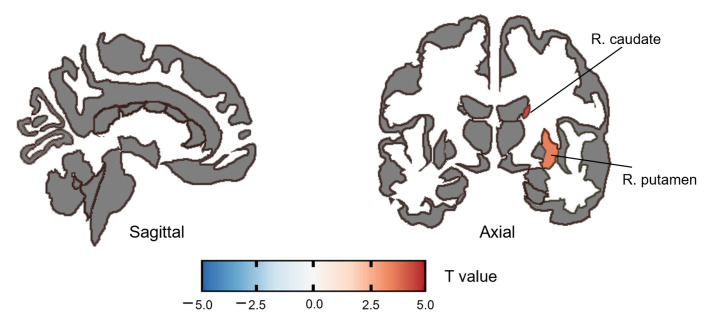
Altered ALFF in the SZ group compared to the HC group. The color bar represents the range of *t* values.

**Figure 2 brainsci-13-01403-f002:**
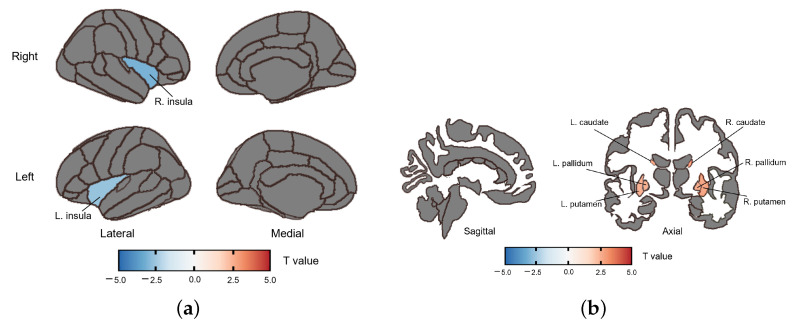
Altered fALFF in the SZ group compared to the HC group. The color bar represents the range of *t* values. (**a**) The fALFF change in the insula. (**b**) The fALFF change in the caudate, putamen and pallidum.

**Figure 3 brainsci-13-01403-f003:**
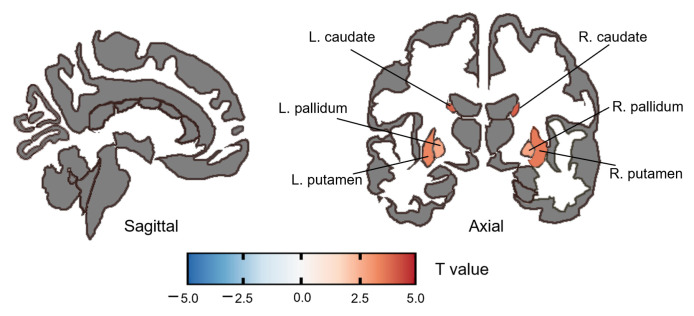
Altered ReHo in the SZ group compared to the HC group. The color bar represents the range of *t* values.

**Figure 4 brainsci-13-01403-f004:**
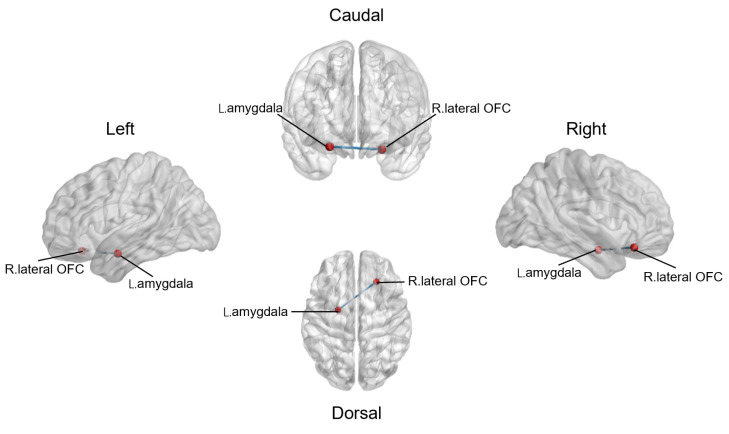
Altered resting-state functional connections between the reward network and the emotional salience network in the SZ group, compared to the HC group; OFC, orbitofrontal cortex.

**Table 1 brainsci-13-01403-t001:** The brain regions in the reward network and the emotional salience network.

Brain Region	Side	Brain Network
caudate	R	reward network
	L	reward network
putamen	R	reward network
	L	reward network
pallidum	R	reward network
	L	reward network
amygdala	R	reward network; emotional salience network
	L	reward network; emotional salience network
caudal anterior cingulate	R	reward network
	L	reward network
lateral OFC	R	reward network; emotional salience network
	L	reward network; emotional salience network
medial OFC	R	reward network; emotional salience network
	L	reward network; emotional salience network
posterior cingulate	R	emotional salience network
	L	emotional salience network
rostral anterior cingulate	R	reward network
	L	reward network
insula	R	reward network; emotional salience network
	L	reward network; emotional salience network

R, right; L, left; OFC, orbitofrontal cortex.

**Table 2 brainsci-13-01403-t002:** Demographic and clinical outcomes of SZ patients and healthy controls.

Items	SZ(n = 30)	HC(n = 30)	$t/\chi^{2}$	*p*
Gender (male/female), n	22/8	17/13	1.83	0.18
Age (years), M(SD)	46.70 (6.47)	47.47 (14.15)	−0.27	0.79
Education (years), M(SD)	12.23 (2.80)	10.50 (4.41)	1.82	0.08
Duration of illness (months), M(SD)	214.12 (104.32)	n/a		
Medication dosage (OL eq.), M(SD)	14.34 (7.44)	n/a		
PANSS, M(SD)				
Positive	14.00 (4.83)	n/a		
Negative	21.63 (4.55)	n/a		
General	34.47 (8.03)	n/a		
Total	70.10 (13.93)	n/a		

*p* values are the differences in demographic and clinical outcomes between two groups on the *t*-test or the chi-square test; $\chi^{2}$ chi-square; M, mean; SD, standard deviation; OL eq., olanzapine equivalents (mg/day); SZ, schizophrenia patients; HC, healthy controls; PANSS, Positive and Negative Symptom Rating Scale.

**Table 3 brainsci-13-01403-t003:** Group differences of ALFF in key brain regions of the reward network and the emotional salience network.

Brain Region	Side	*t*	*p*	$P_{\mathrm{FDR}}$
SZ > HC				
caudate	R	4.40	<0.001	0.001
putamen	R	3.50	<0.001	0.009

SZ, schizophrenia patients; HC, healthy controls; R, right; L, left; SZ > HC, the mean ALFF value of the brain region in SZ was larger than HC. *p* values are the differences in mean ALFF values between two groups on *t*-test. $P_{FDR}$ is *p* value after FDR correction.

**Table 4 brainsci-13-01403-t004:** Group differences of fALFF in key brain regions of the reward network and the emotional salience network.

Brain Region	Side	*t*	*p*	$P_{\mathrm{FDR}}$
SZ > HC				
caudate	R	2.732	0.008	0.038
	L	2.708	0.009	0.038
putamen	R	2.717	0.009	0.038
	L	2.506	0.015	0.038
pallidum	R	2.629	0.011	0.038
	L	2.518	0.015	0.038
SZ < HC				
insula	R	−3.082	0.003	0.038
	L	−2.608	0.012	0.038

SZ, schizophrenia patients; HC, healthy controls; R, right; L, left; SZ > HC/SZ < HC, the mean fALFF value of the brain region in SZ was larger/smaller than HC. *p* values are the differences in mean fALFF values between two groups on *t*-test. $P_{FDR}$ is *p* value after FDR correction.

**Table 5 brainsci-13-01403-t005:** Group differences of ReHo in key brain regions of the reward network and the emotional salience network.

Brain Region	Side	*t*	*p*	$P_{\mathrm{FDR}}$
SZ > HC				
caudate	R	4.048	<0.001	0.002
	L	3.987	<0.001	0.002
putamen	R	3.601	<0.001	0.004
	L	3.393	0.001	0.006
pallidum	R	2.761	0.008	0.030
	L	2.697	0.009	0.030

SZ, schizophrenia patients; HC, healthy controls; R, right; L, left; SZ>HC, the mean ReHo value of the brain region in SZ was larger than HC. *p* values are the differences in mean ReHo values between two groups on *t*-test. $P_{FDR}$ is *p* value after FDR correction.

## Data Availability

The data presented in this study are available on request from the corresponding author, after the acceptance of all the co-authors.

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
