# Peer review of "Deficits in Key Brain Network for Social Interaction in Individuals with Schizophrenia"

_brainsci, 2023, doi:10.3390/brainsci13101403_

Round 1

Reviewer 1 Report

This is an interesting and technically sound neuroimaging study in schizophrenia. The key finding is that resting-state brain function as measured by four different methods is linked to social dysfunctions in schizophrenia, with a special reference to reward processing and saliences. The brain structures emerged from the results are well-known in the literature, including the orbitofrontal cortex, basal ganglia, and the amygdala.

Comments and suggestions:

  1. My main concern is that the authors used multiple types of measurement in a small group of patients and controls. Therefore, the statistical power is low.
  2. The AFNI toolkit should be referred. How did the authors conduct motion correction and temporal layers?
  3. How was the partial volume effect corrected? It is very likely that patients and controls showed substantial brain structural differences.
  4. The measurement of social functions is not clear from the manuscript. What kind of clinical scale was correlated with brain activation?
  5. Please improve the quality of figures 1-3. ROIs are sketchy. Figure 4 is not informative because the brain structures cannot be seen.  
  6. What about the medications (daily dose, type of drugs)? What does it mean that the medication regimen was relatively stable? 

Author Response

Thank you very much for taking the time to review this manuscript. Your valuable comments and suggestions have given us a lot of insight. Please find the detailed responses in the attachment, and the corresponding revisions to the manuscript are given in blue text. The modifications of language are given in green text.

Reviewer 2 Report

Dear authors,

Firstly, I would like to inform that I don´t have any potential conflict of interest neither any other ethical concerns with regards to your paper:

Deficits in Key Brain Network for Social Interaction in Individuals with Schizophrenia.

By Yiwen Wu and Cols.

I have provided my comments as follows:

In this manuscript authors aimed to asses try through magnetic resonance the deficits of brain functioning of patients with schizophrenia in relation to social interaction. Measuring originality poses a challenge for me even in my field of research. Personally, my expectations in neuroimaging studies like this are low; they were promised as a revolution but after many years they have not brought great real clinical changes so the clinical impact and originality are, in my view, average-low however the scientific soundness is quite well. But, apart from that, it was a pleasure to review this paper as the authors have shown a clarity and conciseness that should be appreciated; they present their results in a clearly written and well-organized without superfluous arguments neither unnecessary detours. The information provided is comprehensive and I like the way it is shown. To end I provide some comments with minor mistakes and methodological concerns that I found as follows (most of the latter would become limitations of this research as they are not easy to solve a posteriori):

My major concern about this work is that it seems as if it were part of a larger one (a clinical trial or something like this). If that were the case nothing happens, but the authors should recognize it because it seems so.

 In lines 21 to 24 the authors claim that schizophrenia is characterized by psychotic symptoms and always accompanies social functioning decline. Positive symptoms, such as hallucinations and delusions, and negative symptoms, typified by the loss of motivation, goal-directed behaviors and affects are two main dimensions of SZ symptoms. This sounds too vague and simple for me, so please, add some recent data regarding negative symptoms and expand the information provided.

You should explain better the exclusion criteria B Alcohol abuse as you said: “3 liters or more per drink” I don't quite understand what it means ¿per day?

In line 151 explain briefly what a ROI-to-ROI analysis is and why you chose it.

You need a title for figure 2 and a footer for Tables 1,3,4 and 5 explaining what they mean. Review titles and footer of all the figures and tables.

To end I would like to congratulate the authors for the numerous bibliographic references provided and for the great work done

Yours Sincerely,

The reviewer

Please, check that you have previously cited these acronyms (PALFF, fALFF, ReHo and fc). In the case of the first two, as they are concepts related to magnetic resonance, briefly explain what they are and their importance. In line 107 the acronym TR do not match with repetition time so please review it and change it if necessary. In line 123 I didn’t find this acronym (ANTs) described before

understand the article but, as I am not a native English speaker, nevertheless some expressions and phrases like: “saliently”, “neuroimaging works” or “the findings of this study should be interpreted in light of some limitations” they sound weird to me so I would recommend a linguistic revision of the translation to the authors.

Author Response

(The authors gave the same response as above.)

Reviewer 3 Report

This paper examines functional alterations in brain networks related to social functioning in a small sample (n = 30) of patients with schizophrenia, compared with a matched control group.

Though relatively modest in scope, this study can be viewed as an incremental cognition to the neuroimaging literature in schizophrenia. The authors have used standard imaging protocols, case definitions and symptom rating scales, and their statistical analysis has followed commonly used principles.

The following are aspects of the paper that would benefit from correction or clarification:

1. In the Introduction, the authors' description of the causal links between schizophrenia and social impairment should be clearer. The statement that schizophrenia "always accompanies social functioning decline" is misleading. Rather, the authors should state that schizophrenia is always accompanied by impairments in social behavior and functioning, and should discuss the relationship between symptom dimensions in schizophrenia and the extent and severity of these impairments. The authors have briefly discussed positive and negative symptom dimensions, but have not reviewed their association with social impairment.

2. Likewise, the discussion of the links between cognitive impairment and social functioning in schizophrenia (lines 25-26) is poorly worded. It would be more accurate to state that lower-order cognitive deficits correlate with impairments in social cognition and behaviour, and to provide citations from the literature to this effect. Social cognition deficits in schizophrenia should also be reviewed briefly.

3. The authors' description of the "emotional salience network" is somewhat simplified. The term itself is not frequently used in the literature; rather, there are linked networks for emotional salience and regulation. These involve not just the regions mentioned by the authors, but other cortical structures such as the temporal pole, parietal cortex, middle temporal gyrus, and anterior cingulate cortex. This should be covered in the Introduction.

4. Prior studies of ALFF and related parameters in schizophrenia, in relation to the regions / circuits analyzed in the current study, do exist; it is inaccurate to state that "the functional activity remains unknown" in these regions. It would be useful to cite and discuss some of the recent work in this area (e.g., Cheon et al., 2023; Shao et al., 2022; Qiu et al., 2022).

5. How was the sample size of n = 30 arrived at? Was the study adequately powered to detect meaningful associations / correlations for at least the primary regions of interest? These details should be provided in the methods section.

6. The definition of alcohol abuse provided ("3 litres or more per drink") is not a standard one. 3 litres of any single alcoholic beverage, consumed as one drink, is an extremely high dose. The authors should use a standard definition of alcohol use / problem drinking for a study of this sort.

7. The study subjects were all receiving medication. Given that some previous studies have used drug-naive subjects, how could this have influenced the study results? What are the effects of atypical antipsychotics on the emotional salience and reward networks? Was any method used to adjust for this in data analysis (e.g., conversion of antipsychotic doses to standard values and partial correlation / regression / analysis of covariance?)

8. As mentioned in point 3 above, the emotional salience network consists of several interconnected cortical and subcortical regions. However, this study has focused only on the amygdala, orbitofrontal cortex and insula. What was the rationale for studying these regions and not others of relevance to emotional salience?

9. The authors have used Pearson's r to examine correlations between functional activity measures and PANSS scores. Was this method appropriate for a small sample (n = 30) with a likely non-normal distribution? Would the results have differed if a non-parametric correlation (Spearman, Kendall) was used?

10. The Discussion section should cite and discuss recent similar studies and contrast the current results with those obtained by other researchers (see point 4). Studies published between 2021-23 could be covered here, as the authors' references have cited and discussed earlier studies.

11. Two of the references (36, 37) are incomplete; please provide the year of publication.

A moderate degree of language editing is required in view of several minor errors in grammar, word choice, and sentence construction.

Author Response

(The authors gave the same response as above.)

Round 2

Reviewer 1 Report

Although the authors made substantial revisions in the paper, I still have concerns regarding the statistical power, the lack of correlation with social functions (this finding does not support the hypothesis), and the pure ROI-based approach. The main research question of the study could be answered with more advanced imaging data analyses (e.g., supervised and non-supervised learning models). 

Reviewer 3 Report

The revised manuscript has addressed the concerns raised in my previous report. I have no further major changes to suggest.

Minor language editing may be required in some places.